# NormSoftmax: Normalize the Input of Softmax to Accelerate and Stabilize Training

## Abstract

Softmax is a basic function that normalizes a vector to a probability distribution and is widely used in machine learning, most notably in cross-entropy loss function and dot product attention operations. However, optimization of softmax-based models is sensitive to the input statistics change. We observe that the input of softmax changes significantly during the initial training stage, causing slow and unstable convergence when training the model from scratch. To remedy the optimization difficulty of softmax, we propose a simple yet effective substitution, named NormSoftmax, where the input vector is first normalized to unit variance and then fed to the standard softmax function. Similar to other existing normalization layers in machine learning models, NormSoftmax can stabilize and accelerate the training process, and also increase the robustness of the training procedure against hyperparameters. Experiments on Transformer-based models and convolutional neural networks validate that our proposed NormSoftmax is an effective plug-and-play module to stabilize and speed up the optimization of neural networks with cross-entropy loss or dot-product attention operations.

## 1 Introduction

Softmax is a critical and widely used function in machine learning algorithms, which takes a vector as input and generate a standard simplex. It is usually used to generate a categorical probability distribution. The most notable applications of softmax are cross-entropy loss function for classification tasks and attention map generation in dot product attention operations. By importing the temperature in softmax, we can control the information entropy and sharpness of its output.

However, gradient-based optimization of softmax-based models often suffers from slow and unstable convergence and is sensitive to optimization hyperparameters. Transformer-based models (Vaswani et al., 2017) are known to be hard to optimize. A lot of efforts have been devoted to solving this optimization difficulty (Liu et al., 2020). For instance, Bolya et al. (2022) reports that softmax attention may crash with too many heads and proposes new attention functions. Chen et al. (2021) show that the Vision Transformer's (Dosovitskiy et al., 2021) loss landscape is very sharp, and it requires advanced optimizers to facilitate its training (Foret et al., 2020). Huang et al. (2020) propose a better initialization to improve the Transformer optimization. Xiong et al. (2020) show that the location of layer normalization (LN) has a remarkable impact on the gradients and claim that the Pre-LN Transformer has better training stability.

Among comprehensive reasons for the optimization difficulty of Transformers, cascaded softmax functions are one of them that leads to the training instability. However, limited prior work has discussed the impacts of softmax on optimization. Based on our experimentation, we find that the training difficulty can be attributed to the rapid change in the variance of the softmax inputs and the information entropy of its outputs. In dot-product attention, where the softmax is used to generate weight distribution for key-value pairs, we observe significant statistical fluctuation in softmax inputs. The rapid and extensive variance change in the initial learning stage can lead to unstable training. Moreover, for the softmax used in cross-entropy loss for classification problems, the input of the softmax usually has a lower variance at the initial training stage since the model has less knowledge of the problem (Wei et al., 2022). The model is likely to stay in the low-confidence zone, implying that it is difficult to train (Pearce et al., 2021). We need a specially designed mechanism to push the model out of this low-confidence zone for stable and fast learning.

In the two cases above, the significant change of the softmax input variance is one of the reasons for optimization difficulty. In this paper, we propose NormSoftmax to stabilize and accelerate training by simply re-scaling the softmax inputs, especially in the early stage optimization.

With NormSoftmax, we dynamically calculate vector-specific factors to scale the inputs before being fed to the standard softmax. Specifically, when the input variance is too small, Softmax will generate small gradients that hinder the learning process. In contrast, our proposed NormSoftmax can help re-scale the input distribution such that the information entropy of the output becomes stable without fluctuation during the training process, which boosts and stabilizes the early stage training.

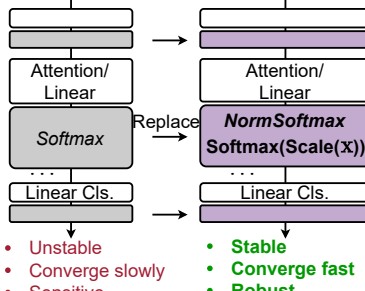

Figure 1: The standard softmax and proposed NormSoftmax.

NormSoftmax shares similar properties with the existing normalization techniques in machine learning. We summarize its advantages below.

- NormSoftmax can re-scale gradients to stabilize the training process, making the training robust to model architectures and optimization recipes (such as optimizers and weight decay schedules).

- NormSoftmax can accelerate the early training stage without hurting the model representability.

- NormSoftmax is an easy-to-use and low-cost module to replace standard softmax. The induced computation and memory cost overhead is negligible.

- NormSoftmax has a regularization effect since the re-scaling can slightly restrict the representation space of the input vectors.

In this paper, we focus on two applications of the softmax functions: (1) the activation function in dot-product attention, and (2) cross-entropy loss of the classification problem. ViT-B with our NormSoftmax shows significantly higher robustness to different head settings, showing an average of **+4.63%** higher test accuracy on CIFAR-10 than its softmax-based counterpart. When training for 100 epochs on ImageNet-1K, ViT with our NormSoftmax can achieve **+0.91%** higher test accuracy over its softmax baseline.

## 2 BACKGROUND

We briefly introduce the softmax function and normalization in machine learning. Then we discuss the two cases we focus on in this paper: softmax in dot product attention and cross entropy loss. Throughout this paper, we use $\mu(\boldsymbol{a}), \sigma(\boldsymbol{a})$ to represent the mean and standard deviation (square root of the variance) of a vector $\boldsymbol{a}$.

### 2.1 SOFTMAX

The standard softmax function $\boldsymbol{z} = \text{softmax}(\boldsymbol{x})$, where $\boldsymbol{x}, \boldsymbol{z} \in \mathbb{R}^n$ is defined by Equation 1.

$$z_i = \frac{e^{x_i}}{\sum_{j=1}^{n} e^{x_j}}, \text{for } i = 1, 2, ..., n \tag{1}$$

The output of softmax can be seen as a categorical probability distribution since $0 < z_i < 1$ and $\sum_i z_i = 1$. Instead of $e$, we can also use a different base in softmax. A temperature parameter $T > 0$ is imported to adjust the base.

$$\text{softmax}_T(\boldsymbol{x}) = \text{softmax}\left(\frac{\boldsymbol{x}}{T}\right) \tag{2}$$

Given the same input vector $\boldsymbol{x}$, the higher temperature smooths the difference of the input vector and generates a probability distribution with high information entropy $H(\boldsymbol{z}) = -\sum_i z_i \log(z_i)$. On the contrary, the lower temperature sharpens the output distribution with low entropy. (Agarwala et al., 2020) claim that the temperature has a crucial impact on the initial learning process.

## 2.2 NORMALIZATION

Normalization layers (Ioffe & Szegedy, 2015; Ba et al., 2016; Ulyanov et al., 2016; Wu & He, 2018) usually normalize the input tensor such that the result has a zero-mean and unit-variance along specific dimensions. We can optionally scale and shift the normalized tensor further. Normalization can accelerate and stabilize the optimization by smoothing the loss landscape (Santurkar et al., 2018; Xu et al., 2019; Bjorck et al., 2018). Hence, normalization allows for a larger learning rate and increases the robustness against hyperparameters. Also, normalization helps generalization since the sharpness of the loss surface is decreased effectively (Lyu et al., 2022). However, we find that normalization is usually used in the intermediate linear transformation layers, and it is rarely applied to the input of softmax functions.

## 2.3 DOT PRODUCT ATTENTION

The scaled dot product attention (Vaswani et al., 2017) is defined by the following equation, where $Q, K, V$ are query, key, and value matrices and $d$ is the dimension of the key vector.

$$\text{Attention}(Q, K, V) = \text{softmax}\left(\frac{QK^T}{\sqrt{d}}\right) V \tag{3}$$

For every query vector, the softmax function calculates the weight for all key-value pairs. The scaling factor of $d^{-1/2}$ is proposed to attempt the normalize the dot product $q^T k$, whose variance is $d$ if the components of $q, k$ are independent random variables with a variance of 0. [1] The scaling factor is applied to address the issue that the variance of dot product $q^T k$ will likely increase as the length of the vector $d$ increases. The large variance makes the gradient of softmax extremely small, thus making the attention-based model hard to train. Based on self-attention, Transformer has achieved great success in many areas, especially natural language processing (Devlin et al., 2019) and computer vision (Khan et al., 2021).

## 2.4 CROSS-ENTROPY LOSS

Another important application of softmax is in classification problems, where minimizing the cross-entropy loss is equivalent to maximizing the likelihood. The cross-entropy function takes the estimated probability distribution $q = \text{softmax}(x)$ and the true probability distribution $p$ as input and compute the result by $H(p, q) = -\sum_i p_i \log q_i$. $x$ is the predicted logits, usually generated by a classification model. $x$ can be any vector in $\mathbb{R}^K$ without restrictions, where $K$ is the number of classes.

## 3 METHOD

In this section, we first analyze the behavior of the softmax input during the training process. Further, we define NormSoftmax and discuss its advantage in the two cases.

### 3.1 UNDERSTANDING THE BEHAVIOR OF SOFTMAX IN DIFFERENT TRAINING STAGES

We split the whole training process into three stages.

- **Initial stage.** Starting from scratch, we usually need a careful design for initialization and hyperparameters. The training may fail due to exploding or vanishing gradients. For example, we usually apply learning rate warmup during this stage.

- **Intermediate stage.** Once the parameters are sufficiently warmed up, it is relatively easy and stable to explore the solution space with a higher learning rate.

- **Final stage.** The model attempts to converge to a local minimum with a lower learning rate.

The initial stage is the most unstable one among these three stages. If we do not encounter severe issues in the initial stage, it is likely that we can proceed to the final results.

---

[1]Please refer to Footnote 4 of the original paper (Vaswani et al., 2017).

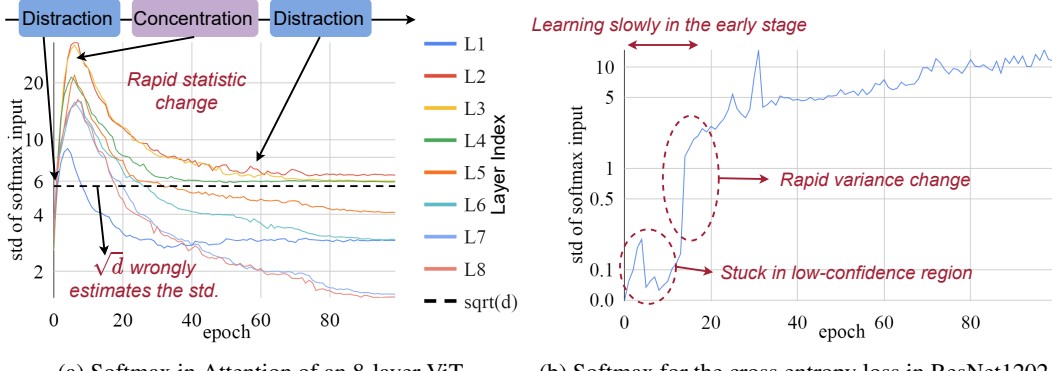

(a) Softmax in Attention of an 8-layer ViT    (b) Softmax for the cross entropy loss in ResNet1202

Figure 2: The standard deviation of softmax input has significant change during the training process. Note that the vertical axes are in the logarithmic scale.

Figure 2 illustrates two cases where we observe a significant change in the softmax input variance throughout the training process. We discuss the behavior of the softmax function in these two cases.

**Softmax in dot product attention.** The input variance of softmax is related to the attended region of this attention layer. With a high variance on the input, the output of softmax has a low information entropy, meaning that only few keys are attended. On the contrary, the smaller input variance implies that much more keys are attended. Figure 2a demonstrates that the standard deviation of softmax input (before scaled with $\sqrt{d}$) increases rapidly at the initial stage and then decreases gradually during the intermediate and final stages. In the beginning, the vision Transformer (Dosovitskiy et al., 2021) attempts to attend to almost all the keys since the initialized model has not learned how to extract features and is still in the exploration stage. During training, different keys are learned at different paces. Keys with smaller semantic distances from nearby queries are much easier to learn. This imbalanced learning pace among keys drives the model to shrink its receptive field and focus on a small region. That is why the variance of softmax input increases rapidly at the initial stage. Afterwards, as training proceeds, Transformer will explore the query-key pairs with longer semantic distance, implied by a gradual reduction in the variance of softmax input. We name the effect "distraction-concentration-distraction".

Further, different layers have different input variance. Except for the first layer (L1), the lower layer has a high variance than the deeper layers. In the early layers, only a small region is attended to. As depth grows, the model attends to a larger region, which is similar to the trend of receptive fields in convolutional neural networks. We also draw the line of $\sqrt{d}$ as a reference in Figure 2a, which is the scaling factor used to normalize the softmax input. Since the *input variance has a significant change across different layers and training steps, this constant scaling factor of $\sqrt{d}$ might not be the most suitable value* to normalize the softmax input (Lee et al., 2021).

**Softmax in cross-entropy loss function.** For classification model training, the softmax may only appear in the cross-entropy loss function, e.g., ResNet. However, the gradients through softmax can have a large impact on the model training. To verify this claim, we train a deep ResNet1202 (He et al., 2016) on CIFAR10 dataset (Krizhevsky et al., 2009) and plot the standard deviation of the softmax input in Figure 2b. At the initial stage, the standard deviation is relatively low since the model is less confident, and the predicted distribution is similar to a uniform distribution. There is a leap at the 15-th epoch, where the standard deviation increases from 0.14 to 1.31. Afterward, the standard deviation of the softmax input gradually increases, and the information entropy of the softmax output becomes smaller since the model becomes more confident in predictions as training continues. Hence, we conclude that the variance of softmax input experiences rapid and huge change during training, especially in the initial stage, which explains why training from scratch is difficult.

Moreover, the observed trend in Figure 2 is averaged on all the data points. Actually, different data points or training examples are in different learning stages. For instance, in the image classification problem, a clear image can easily escape the less-confident zone quickly and stably, while a vague

image may be stuck in this zone where the softmax output has a high information entropy. Therefore, this training difficulty needs to be tackled in a data-specific fashion.

## 3.2 THE PROPOSED NORMSOFTMAX

We propose NormSoftmax as a substitution for softmax, as defined below,

$$\text{NormSoftmax}(\boldsymbol{x}, \gamma) = \text{softmax}\left(\frac{\boldsymbol{x} - \mu(\boldsymbol{x})\mathbf{1}}{\min(\sigma(\boldsymbol{x}), \gamma)}\right) \tag{4}$$

$$= \text{softmax}\left(\frac{\boldsymbol{x}}{\min(\sigma(\boldsymbol{x}), \gamma)}\right) \tag{5}$$

$$= \text{softmax}\left(\frac{\boldsymbol{x}/\gamma}{\min(\sigma(\boldsymbol{x}/\gamma), 1)}\right) \tag{6}$$

where $\gamma > 0$ is a pre-defined scalar. Namely, we define the temperature $T = \min(\sigma(\boldsymbol{x}), \gamma)$ in NormSoftmax. Since softmax is invariant under translation by the same value, Equations 4 and 5 are equivalent. We do not need to shift the input vector to zero-mean. If the standard derivation is smaller than the threshold $\sigma(\boldsymbol{x}) \leq \gamma$, NormSoftmax will normalize the input vector to obtain a unit-variance vector before applying the standard softmax function. If $\sigma(\boldsymbol{x}) > \gamma$, we use the temperature $T = \gamma$. If $\gamma = +\infty$, then we will always normalize the input vector. The temperature is dynamically calculated *per vector*. For a batch of vectors, each one has its individual temperature.

**Lemma 1** *Given* $\boldsymbol{y} = \frac{\boldsymbol{x}}{\sigma(\boldsymbol{x})}, \boldsymbol{z} = softmax(\boldsymbol{y}), \frac{\partial l}{\partial \boldsymbol{z}} \in \mathbb{R}^n$, *we have*

$$\mu\left(\frac{\partial l}{\partial \boldsymbol{x}}\right) = \mu\left(\frac{\partial l}{\partial \boldsymbol{y}}\right) = 0, \sigma\left(\frac{\partial l}{\partial \boldsymbol{x}}\right) \leq \frac{\sigma\left(\frac{\partial l}{\partial \boldsymbol{y}}\right)}{\sigma(\boldsymbol{x})}, \left\|\frac{\partial l}{\partial \boldsymbol{x}}\right\|_2 \leq \frac{\left\|\frac{\partial l}{\partial \boldsymbol{y}}\right\|_2}{\sigma(\boldsymbol{x})} \tag{7}$$

**Lemma 2** *Given* $\boldsymbol{x}_2 = k\boldsymbol{x}_1, \boldsymbol{z}_1 = softmax(\boldsymbol{x}_1/\sigma(\boldsymbol{x}_1)), \boldsymbol{z}_2 = softmax(\boldsymbol{x}_2/\sigma(\boldsymbol{x}_2)), \frac{\partial l}{\partial \boldsymbol{z}_1} = \frac{\partial l}{\partial \boldsymbol{z}_2} \in \mathbb{R}^n$, *we have* $\boldsymbol{z}_1 = \boldsymbol{z}_2, \frac{\partial l}{\partial \boldsymbol{z}_1} = \frac{\partial l}{\partial \boldsymbol{z}_2}, \frac{\partial l}{\partial \boldsymbol{x}_2} = \frac{1}{k}\frac{\partial l}{\partial \boldsymbol{x}_1}$.

Similar to the theorem in Xu et al. (2019), we refer to gradient re-centering and re-scaling as gradient normalization. Similar to Theorem 4.1 in Santurkar et al. (2018), we demonstrate that the normalization improves the Lipschitz continuity, indicated by the gradient magnitudes. Specifically, $\left\|\frac{\partial l}{\partial \boldsymbol{x}}\right\|_2$ and $\left\|\frac{\partial l}{\partial \boldsymbol{y}}\right\|_2$ can be treated as the continuity of the loss function Given two input vectors $\boldsymbol{x}_1, \boldsymbol{x}_2$ and $\sigma(\boldsymbol{x}_1) < \sigma(\boldsymbol{x}_2) < \gamma$, NormSoftmax apply a different temperature on them, paying much attention to the low-variance vector $\boldsymbol{x}_1$. In Lemma 2, if $\sigma(\boldsymbol{x}_1) = 1$, then $k = \sigma(\boldsymbol{x}_2)$, it is clear that the gradients are rescaled by its variance $\frac{\partial l}{\partial \boldsymbol{x}_2} = \frac{1}{\sigma(\boldsymbol{x}_2)}\frac{\partial l}{\partial \boldsymbol{x}_1}$.

Logit Normalization (LogitNorm) (Kornblith et al., 2021; Wei et al., 2022) uses the $\ell_2$ norm to normalize the input vector, as shown in the equation below, where $\tau$ is the temperature parameter modulating the magnitude of the logits.

$$\text{LogitNorm}(\boldsymbol{x}, \tau) = \text{softmax}\left(\frac{\boldsymbol{x}}{\tau\|\boldsymbol{x}\|_2}\right) \tag{8}$$

LogitNorm is proposed to mitigate overconfidence when cross entropy loss is used. It is questionable that LogitNorm does not shift the input vector to zero-mean, since the mean has an impact on the $\ell_2$ norm. Unlike the standard softmax function, LogitNorm is not invariant under translation $\text{LogitNorm}(\boldsymbol{x}, \tau) \neq \text{LogitNorm}(\boldsymbol{x} + c\mathbf{1}, \tau)$, where $c$ is a constant.

We demonstrate the relationship between LogitNorm and NormSoftmax in the following equation.

$$\text{NormSoftmax}(\boldsymbol{x}, \gamma = +\infty) = \text{softmax}\left(\sqrt{n}\frac{\boldsymbol{x} - \mu(\boldsymbol{x})\mathbf{1}}{\|\boldsymbol{x} - \mu(\boldsymbol{x})\mathbf{1}\|_2}\right) = \text{LogitNorm}(\boldsymbol{x} - \mu(\boldsymbol{x})\mathbf{1}, \tau = n^{-1/2}) \tag{9}$$

NormSoftmax first shifts the input vector to zero-mean and normalizes the shifted vector by its $\ell_2$ norm. Hence, NormSoftmax keeps the invariance under translation and can be reduced to Logit-Norm with input shifting.

### 3.3 EFFECTS OF NORMSOFTMAX IN THREE STAGES

**NormSoftmax can accelerate and stabilize training in the initial stage.** With NormSoftmax, the standard deviation of the softmax input is at least 1 since $\sigma\left(\frac{x}{\min(\sigma(x),\gamma)}\right) \geq 1$. If the softmax input has low variance, which is common in the initial stage for both two applications, we use a self-adapted low temperature to magnify the slight difference. The normalization can help the model escape the zone with high information entropy quickly and stably since we normalize the gradients as demonstrated in Theorem 1. Specifically, for attention layers, NormSoftmax can accelerate the transition from distraction to concentration.

**NormSoftmax can regularize training in the intermediate and final stages.** For cross entropy loss, NormSoftmax increases the temperature of the softmax in the training process since the variance of the softmax input will increase gradually. The high temperature can regularize the training. For attention layers, NormSoftmax regularizes the softmax input, thus restricting the attentive areas. NormSoftmax encourages the attention to have a slight change on the attended regions, which can be treated as an inductive bias we add.

However, we find that by simply normalizing inputs to unit-variance vectors, i.e., $x/\sigma(x)$, the training process can be impeded due to overly restricted representation space. Our variance clipping technique can effectively solve this issue with a pre-defined threshold $\gamma$.

## 4 EXPERIMENTS

Detailed experiment settings can be found in Appendix. Since we primarily use the cosine annealing learning rate scheduler, we train from scratch when the total number of epochs is different. Namely, the learning rate schedule is updated according to the number of epochs.

### 4.1 DOT-PRODUCT ATTENTION

**Settings.** We train the vision transformer (ViT) on the CIFAR10 dataset from scratch with AdamW (Loshchilov & Hutter, 2019) optimizer for 100 epochs (50,000 iterations with 100 mini-batch size). The resolution of an input image is $3 \times 32 \times 32$, and the patch size is 4. The hidden size, MLP size, number of heads, and the dimension of heads are 256, 1024, 8, and 32, respectively. We discard the original scaling factor $\sqrt{d}$ and replace the standard softmax with NormSoftmax, setting $\gamma = \sqrt{d}$ or $\gamma = +\infty$. The learning rate is linearly increased with 5 warmup epochs and then decays with the cosine annealing scheduler (Loshchilov & Hutter, 2017). Following the original ViT paper (Dosovitskiy et al., 2021), the learning rate is $1e-3$, and a strong weight decay $1e-1$ is applied. We also enable label smoothing (Szegedy et al., 2016) and strong data augmentation (random erasing (Zhong et al., 2020), mixup (Zhang et al., 2018), cutmix (Yun et al., 2019), and TrivialAugment (Müller & Hutter, 2021)). Figure 3 shows the detailed results.

**Acceleration.** Figure 3a demonstrates the result with different training epochs. When the number of epochs is small, NormSoftmax can achieve a better test accuracy than the standard softmax function. When we train with more iterations, NormSoftmax performs similarly to the baseline. The larger pre-defined threshold $\gamma$ may translate into a higher acceleration, which pushes the model to escape the initial distraction stage more quickly. That is why `NSM-inf` performs better than `NSM-sqrtd` when the number of epochs is small. However, the normalization has a strict regularization effect on the softmax input, which impedes the training in the intermediate and final stages. Hence, the `NSM-inf` is exceeded by `NSM-sqrtd` when the number of training iterations is large. In short, NormSoftmax can accelerate the training process without sacrificing the representation ability, similar to curriculum learning (Wu et al., 2020).

**Stabilization.** We investigate the role of each component in the training recipe, with results listed in Figure 3b. We conduct ablation studies by removing (1) learning rate warmup, (2) weight decay, (3) label smoothing, and (4) strong data augmentation (random erasing, mixup, cutmix, and TrivialAugment) separately. We also replace the default AdamW optimizer with stochastic gradient descent with momentum (SGDM). The results indicate that the techniques above are critical to Transformer no matter what softmax function we use. However, Transformer with standard softmax and scaling factor $d^{-1/2}$ is more sensitive to the training techniques that are related to optimization.

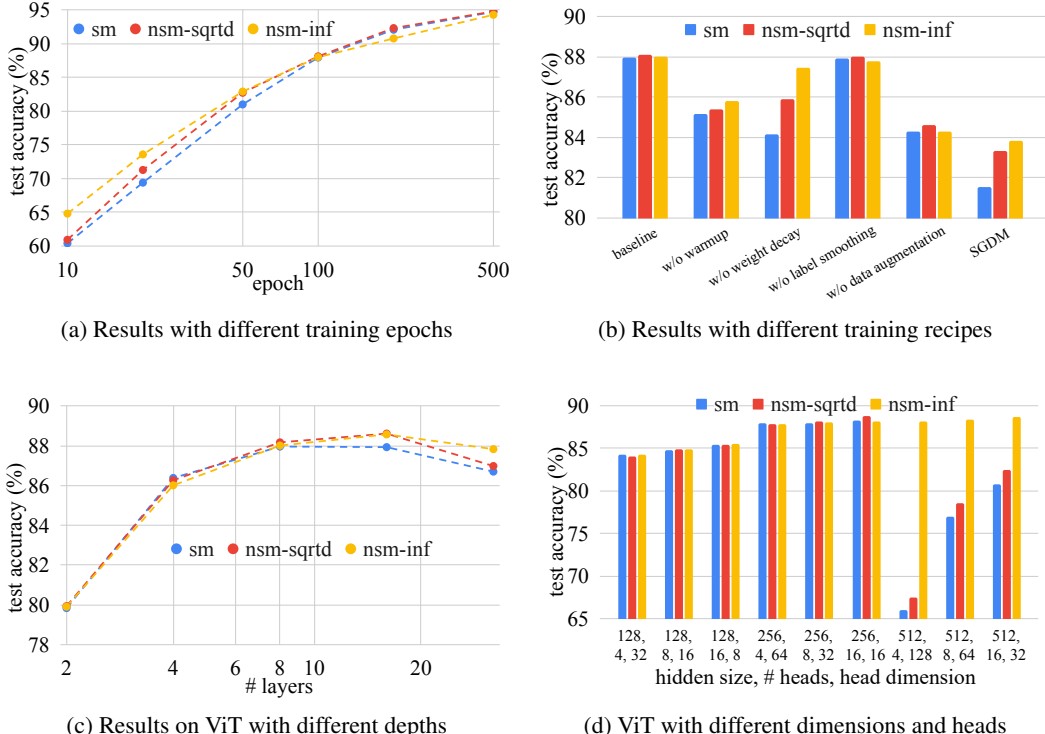

(a) Results with different training epochs

(b) Results with different training recipes

(c) Results on ViT with different depths

(d) ViT with different dimensions and heads

Figure 3: The test accuracy of a ViT on CIFAR10 dataset with different settings. `sm` and `nsm` are short for softmax and NormSoftmax. `sqrtd` and `inf` are the value of $\gamma$ in NormSoftmax.

Without weight decay, the test accuracy of the baseline degrades from $87.96\%$ to $84.15\%$, while the `NSM-inf` has a small drop from $88.01\%$ to $87.46\%$. They share the same robustness against the training recipe for data augmentation.

We also alter the hyperparameters of the ViT and list the results in Figures 3c and 3d. The three methods share similar results when the depth or the hidden dimension is small. However, large depth and hidden dimension impose a challenge for training. `NSM-inf` is more robust and provides much better results than the standard softmax.

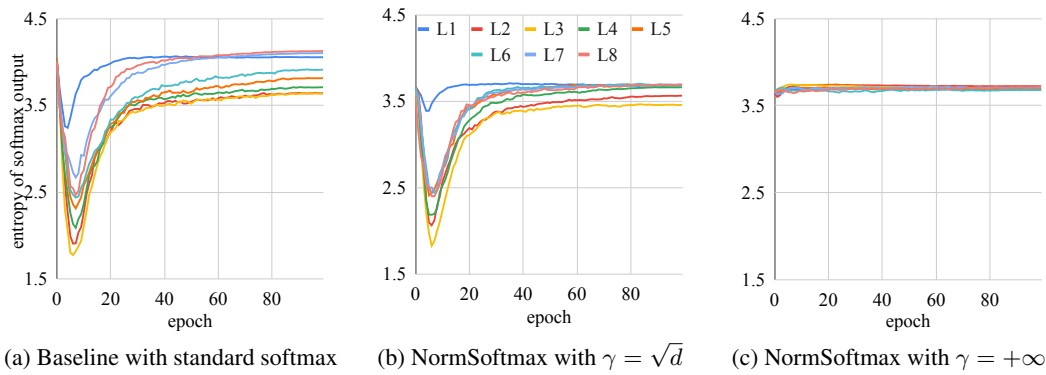

(a) Baseline with standard softmax    (b) NormSoftmax with $\gamma = \sqrt{d}$    (c) NormSoftmax with $\gamma = +\infty$

Figure 4: NormSoftmax reduces the information entropy of softmax output in the 8-layer ViT.

We plot the information entropy of softmax output in this ViT in Figure 4. In baseline, the large standard deviation of softmax input induces small information entropy in the softmax output. The "distraction-concentration-distraction" effect is visualized in Figure 4a. The attended areas of Transformer undergo significant change during the training process. On the other extreme, `NSM-inf`

generates the output, whose information entropy only slightly fluctuates since the input variance is always 1. The small change in the entropy of the output contributes to the acceleration and stability of the initial training process. The `NSM-sqrtd` is an interpolation between these two extremes.

**Variants of NormSoftmax.** We define several variants of NormSoftmax. (1) Adding learnable parameters for affine transformation defined by $f_1(\boldsymbol{x}) = \text{softmax}\left(w\boldsymbol{x}/\sigma(\boldsymbol{x})\right)$, where $w \in \mathbb{R}$ is a learnable parameter. (2) Inverting the NormSoftmax defined by $f_2(\boldsymbol{x}, \gamma) = \text{softmax}\left(\frac{\boldsymbol{x}}{\max(\sigma(\boldsymbol{x}), \gamma)}\right)$ (3) NormSoftmax with different $\gamma$ values, (4) Logit Normalization.

| baseline | $f_1$ | $f_2, \gamma = \sqrt{d}$ | nsm-1 | nsm-$\sqrt{d}/2$ | nsm-$\sqrt{d}$ | nsm-2$\sqrt{d}$ | nsm-$\infty$ | logitnorm |
|---|---|---|---|---|---|---|---|---|
| 87.96 | 87.97 | 86.99 | 81.04 | 87.16 | 88.13 | 88.21 | 88.01 | 87.72 |

Table 1: Train a ViT on CIFAR10 with different NormSoftmax variants.

For the first variant $f_1$, we find that the learnable parameter is not necessary. Indicated by (Xu et al., 2019), the learnable weight may induce the risk of overfitting. The second variant $f_2$ has an inverse clipping and is worse than our proposed NormSoftmax. It cannot accelerate the training since it encourages distraction in the Vision Transformer. In the third variant, the $\gamma$ is a critical hyperparameter. $\gamma = 1, \sqrt{d}/2$ is too small, and the input vector is not effectively scaled. $\gamma = \sqrt{d}, 2\sqrt{d}$ obtains similar result. For Logit Normalizaiton, we sweep the parameter of temperature $\tau$, and find that $\tau = n^{-1/2}$ is almost the best one. With this temperature, the only difference between LogitNorm and NormSoftmax is whether the input is shifted to zero-mean, as shown in Equation equation 9. The accuracy with this temperature is 87.72%, which is even worse than the baseline, demonstrating the importance of shifting.

**Other benchmarks.** We follow the reference implementation provided by PyTorch (Paszke et al., 2019) to train different ViTs on ImageNet (Deng et al., 2009) from scratch. Strong data augmentations and many techniques are adopted. [2] Results are listed in Table 3. NormSoftmax can achieve better performance with a small number of epochs and similar performance with 300-epoch training.

| | 100 epochs | | | 300 epochs | | |
|---|---|---|---|---|---|---|
| | sm | nsm-sqrtd | nsm-inf | sm | nsm-sqrtd | nsm-inf |
| ViT-B-32 | 71.40 | 72.01 | **72.64** | 75.91 | 75.92 | **75.95** |
| ViT-L-32 | 72.54 | **73.49** | 73.46 | 76.97 | **77.01** | 76.92 |
| ViT-B-16 | 76.52 | 77.01 | **77.10** | 81.07 | 81.05 | **81.09** |
| Swin-T | 76.91 | **77.50** | 77.42 | 81.41 | **81.52** | 81.45 |

Table 2: Test accuracy of three ViT variants trained on ImageNet-1K from scratch.

We conduct experiments on machine translation with Transformers following the settings in (Xu et al., 2019). The benchmarks are WMT English-German Translation (en-de), IWSLT 2014 German-English Translation (de-en), and IWSLT 2015 English-Vietmanese Translation (en-vi) (Cettolo et al., 2015). We replace the standard softmax function in both encoder and decoder with our proposed NormSoftmax. The evaluation metric is BLEU (Papineni et al., 2002). Similar to the results in computer vision, NormSoftmax can also accelerate the learning process in natural language processing.

## 4.2 CROSS ENTROPY LOSS OF THE CLASSIFICATION PROBLEM.

We follow the example in the JAX framework (Bradbury et al., 2018) to train ResNets (He et al., 2016) on ImageNet, which has 1,000 classes and about 1.3 million training images. [3] We use SGDM with linear warmup and cosine annealing learning rate scheduler, accompanied by a large

---

[2] The implementation is at this link.

[3] The implementation is available at this link.

| | 45k steps | | | 90k steps | | |
|---|---|---|---|---|---|---|
| | sm | nsm-sqrtd | nsm-inf | sm | nsm-sqrtd | nsm-inf |
| en-de | 24.2 | 25.6 | 25.5 | 28.3 | 28.3 | 28.2 |
| de-en | 30.1 | 30.9 | 31.5 | 35.4 | 35.5 | 35.4 |
| en-vi | 26.7 | 27.1 | 27.2 | 31.2 | 31.3 | 31.4 |

Table 3: The BLEU on three machine translation benchmarks with Transformers.

mini-batch size of 8,192 and a large learning rate of 3.2. We only enable the horizontal flip and input normalization as data augmentation techniques. We set $\gamma$ as 1 and $+\infty$ in NormSoftmax for the cross entropy loss since 1 is the temperature in the baseline. We also apply Logit Normalization and compare it with our proposed method. For Logit Normalization, we conduct a grid search to find the optimal hyperparameter of temperature.

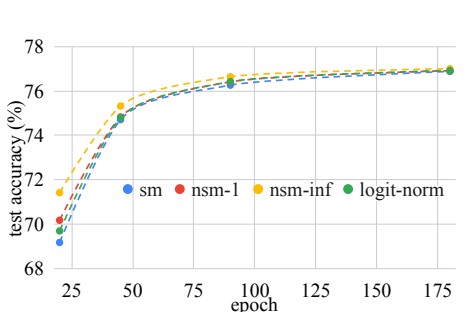

| # epochs | | 20 | 45 | 90 | 180 |
|---|---|---|---|---|---|
| R50 | sm | 69.18 | 74.71 | 76.26 | 76.88 |
| | nsm-1 | 70.18 | 74.83 | 76.41 | 76.91 |
| | nsm-inf | 71.42 | 75.32 | 76.64 | 77.17 |
| | lognorm | 69.70 | 74.79 | 76.42 | 76.92 |
| R101 | sm | 72.44 | 76.21 | 77.79 | 77.99 |
| | nsm-1 | 72.48 | 76.18 | 77.80 | 78.08 |
| | nsm-inf | 72.69 | 76.56 | 78.00 | 78.13 |
| | lognorm | 72.50 | 76.19 | 77.82 | 78.01 |
| R152 | sm | 72.79 | 76.78 | 78.18 | 78.51 |
| | nsm-1 | 72.80 | 76.85 | 78.32 | 78.50 |
| | nsm-inf | 72.60 | 76.73 | 78.17 | 78.44 |
| | lognorm | 72.75 | 76.81 | 78.15 | 78.40 |

Figure 5: Test accuracy of ResNet50 ImageNet-1K with different training epochs

Table 4: Test accuracy on ImageNet with ResNets. *lognorm* is short for Logit Normalization.

Figure 5 and Table 4 show that the NormSoftmax can boost the initial training. With 20 epochs, `NSM-inf` can achieve the test accuracy of 71.42% while the baseline with standard softmax obtains 69.18%. With sufficiently long epochs, `SM` and `NSM` achieve similar test accuracy, implying that `NSM` does not impede the representation learning ability of the models. With larger pre-defined threshold $\gamma$, `NSM-inf` is faster than `NSM-1`. NormSoftmax achieves better results than Logit Normalization, demonstrating that it is necessary to shift the input vector before applying Logit Normalization.

## 5 CONCLUSION

We deeply investigate the behavior of softmax in neural network training and discuss its impacts on training stability and convergence. We find that one of the reasons for the optimization difficulty is the significant change in the variance of softmax input during the early training process. To remedy the optimization difficulty of softmax, we propose a simple yet effective substitution, named NormSoftmax, where the input vectors are first re-scaled by dynamically calculated vector-specific factors and then fed to the standard softmax function. Similar to other existing normalization layers in machine learning models, NormSoftmax can stabilize and accelerate the training process and also increase the robustness of the training procedure to hyperparameters. Experiments on Transformers in computer vision and natural language processing benchmarks validate that our proposed Norm-Softmax is an effective plug-and-play module to stabilize and speed up the optimization of neural networks with cross-entropy loss or dot-product attention operations.

**Limitations.** Transformer-based models are data hungry and its performance is limited given small or intermediate training data. Their representation ability can be unleashed with larger datasets. In this work, we only present the results on small- or mid- size datasets. The training behavior on larger datasets may be different from what we observe on smaller ones.

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

## A    PROOF OF LEMMAS

### A.1    LEMMA 1

Let $\boldsymbol{y} = \frac{\boldsymbol{x}}{\sigma(\boldsymbol{x})}, \boldsymbol{z} = \mathrm{softmax}(\boldsymbol{y})$. Following the chain rules, we have

$$\frac{\partial l}{\partial \boldsymbol{y}} = (\mathrm{diag}(\boldsymbol{z}) - \boldsymbol{z}\boldsymbol{z}^T)\frac{\partial l}{\partial \boldsymbol{z}} \tag{10}$$

$$\frac{\partial l}{\partial \boldsymbol{x}} = \frac{1}{\sigma(\boldsymbol{x})}\left(\boldsymbol{I} - \frac{\boldsymbol{x}\boldsymbol{x}^T - \mu(\boldsymbol{x})\mathbf{1}\boldsymbol{x}^T}{n\sigma^2(\boldsymbol{x})}\right)\frac{\partial l}{\partial \boldsymbol{y}} = \frac{1}{\sigma(\boldsymbol{x})}\left(\boldsymbol{I} - \frac{\boldsymbol{y}\boldsymbol{y}^T - \mu(\boldsymbol{y})\mathbf{1}\boldsymbol{y}^T}{n}\right)\frac{\partial l}{\partial \boldsymbol{y}} \tag{11}$$

With $\mathbf{1}^T\boldsymbol{z} = 1$, we can verify that

$$\mathbf{1}^T\frac{\partial l}{\partial \boldsymbol{y}} = (\mathbf{1}^T\mathrm{diag}(\boldsymbol{z}) - \mathbf{1}^T\boldsymbol{z}\boldsymbol{z}^T)\frac{\partial l}{\partial \boldsymbol{z}} \tag{12}$$

$$= (\boldsymbol{z}^T - \boldsymbol{z}^T)\frac{\partial l}{\partial \boldsymbol{z}} \tag{13}$$

$$= 0 \tag{14}$$

With $n\mu(\boldsymbol{x}) = \mathbf{1}^T\boldsymbol{x}$, we can prove that

$$\mathbf{1}^T\frac{\partial l}{\partial \boldsymbol{x}} = \frac{1}{\sigma(\boldsymbol{x})}\left(\mathbf{1}^T\boldsymbol{I} - \frac{\mathbf{1}^T\boldsymbol{x}\boldsymbol{x}^T - \mu(\boldsymbol{x})\mathbf{1}^T\mathbf{1}\boldsymbol{x}^T}{n\sigma^2(\boldsymbol{x})}\right)\frac{\partial l}{\partial \boldsymbol{y}} \tag{15}$$

$$= \frac{1}{\sigma(\boldsymbol{x})}\left(\mathbf{1}^T - \frac{n\mu(\boldsymbol{x})\boldsymbol{x}^T - n\mu(\boldsymbol{x})\boldsymbol{x}^T}{n\sigma^2(\boldsymbol{x})}\right)\frac{\partial l}{\partial \boldsymbol{y}} \tag{16}$$

$$= \frac{1}{\sigma(\boldsymbol{x})}\mathbf{1}^T\frac{\partial l}{\partial \boldsymbol{y}} \tag{17}$$

$$= 0 \tag{18}$$

Hence, we prove that $\mu\left(\frac{\partial l}{\partial \boldsymbol{x}}\right) = \mu\left(\frac{\partial l}{\partial \boldsymbol{y}}\right) = 0$. We then prove the scaling for $L^2$ norm by leveraging the definition of variance $\sigma^2(\boldsymbol{a}) = \|\boldsymbol{a}\|_2^2/n - \mu^2(\boldsymbol{a})$.

$$\left\|\frac{\partial l}{\partial \boldsymbol{x}}\right\|_2^2 = \left(\frac{\partial l}{\partial \boldsymbol{x}}\right)^T\frac{\partial l}{\partial \boldsymbol{x}} \tag{19}$$

$$= \frac{1}{\sigma^2(\boldsymbol{x})}\left(\frac{\partial l}{\partial \boldsymbol{y}}\right)^T\left(\boldsymbol{I} - \frac{\boldsymbol{x}\boldsymbol{x}^T - \mu(\boldsymbol{x})\boldsymbol{x}\mathbf{1}^T}{n\sigma^2(\boldsymbol{x})}\right)\left(\boldsymbol{I} - \frac{\boldsymbol{x}\boldsymbol{x}^T - \mu(\boldsymbol{x})\mathbf{1}\boldsymbol{x}^T}{n\sigma^2(\boldsymbol{x})}\right)\frac{\partial l}{\partial \boldsymbol{y}} \tag{20}$$

$$= \frac{1}{\sigma^2(\boldsymbol{x})}\left(\frac{\partial l}{\partial \boldsymbol{y}}\right)^T\left(\boldsymbol{I} - \frac{\boldsymbol{x}\boldsymbol{x}^T}{n\sigma^2(\boldsymbol{x})} + \frac{\mu(\boldsymbol{x})\mathbf{1}\boldsymbol{x}^T + \mu(\boldsymbol{x})\boldsymbol{x}\mathbf{1}^T}{n\sigma^2(\boldsymbol{x})}\right)\frac{\partial l}{\partial \boldsymbol{y}} \tag{21}$$

$$= \frac{1}{\sigma^2(\boldsymbol{x})}\left(\left\|\frac{\partial l}{\partial \boldsymbol{y}}\right\|_2^2 - \left(\frac{\partial l}{\partial \boldsymbol{y}}\right)^T\frac{\boldsymbol{x}\boldsymbol{x}^T}{n\sigma^2(\boldsymbol{x})}\frac{\partial l}{\partial \boldsymbol{y}} + \frac{\mu(\boldsymbol{x})\left(\frac{\partial l}{\partial \boldsymbol{y}}\right)^T\mathbf{1}\boldsymbol{x}^T\frac{\partial l}{\partial \boldsymbol{y}} + \mu(\boldsymbol{x})\left(\frac{\partial l}{\partial \boldsymbol{y}}\right)^T\boldsymbol{x}\mathbf{1}^T\frac{\partial l}{\partial \boldsymbol{y}}}{n\sigma^2(\boldsymbol{x})}\right) \tag{22}$$

$$= \frac{1}{\sigma^2(\boldsymbol{x})}\left(\left\|\frac{\partial l}{\partial \boldsymbol{y}}\right\|_2^2 - \left(\frac{\partial l}{\partial \boldsymbol{y}}\right)^T\frac{\boldsymbol{x}\boldsymbol{x}^T}{n\sigma^2(\boldsymbol{x})}\frac{\partial l}{\partial \boldsymbol{y}}\right) \tag{23}$$

$$= \frac{1}{\sigma^2(\boldsymbol{x})}\left\|\frac{\partial l}{\partial \boldsymbol{y}}\right\|_2^2 - \frac{1}{n\sigma^4(\boldsymbol{x})}\left(\boldsymbol{x}^T\frac{\partial l}{\partial \boldsymbol{y}}\right)^2 \tag{24}$$

$$\leq \frac{1}{\sigma^2(\boldsymbol{x})}\left\|\frac{\partial l}{\partial \boldsymbol{y}}\right\|_2^2 \tag{25}$$

From the definition of variance, we obtain that

$$\sigma^2\left(\frac{\partial l}{\partial \boldsymbol{x}}\right) = \left\|\frac{\partial l}{\partial \boldsymbol{x}}\right\|_2^2/n \leq \frac{1}{\sigma^2(\boldsymbol{x})}\left\|\frac{\partial l}{\partial \boldsymbol{y}}\right\|_2^2/n = \frac{1}{\sigma^2(\boldsymbol{x})}\sigma^2\left(\frac{\partial l}{\partial \boldsymbol{y}}\right) \tag{26}$$

Above all, we prove that

$$\left\|\frac{\partial l}{\partial \boldsymbol{x}}\right\|_2 \leq \frac{\left\|\frac{\partial l}{\partial \boldsymbol{y}}\right\|_2}{\sigma(\boldsymbol{x})}, \sigma\left(\frac{\partial l}{\partial \boldsymbol{x}}\right) \leq \frac{\sigma\left(\frac{\partial l}{\partial \boldsymbol{y}}\right)}{\sigma(\boldsymbol{x})} \tag{27}$$

## A.2  LEMMA 2

Given $\boldsymbol{x}_2 = k\boldsymbol{x}_1$, we have $\boldsymbol{x}_1/\sigma(\boldsymbol{x}_1) = \boldsymbol{x}_2/\sigma(\boldsymbol{x}_2)$. Thus, we obtain that $\boldsymbol{z}_1 = \text{softmax}(\boldsymbol{x}_1/\sigma(\boldsymbol{x}_1)) = \boldsymbol{z}_2 = \text{softmax}(\boldsymbol{x}_2/\sigma(\boldsymbol{x}_2))$. Now that $\boldsymbol{z}_1 = \boldsymbol{z}_2$, we can obtain the same loss $f(\boldsymbol{z}_1) = f(\boldsymbol{z}_2)$ and generate the same gradient $\frac{\partial l}{\partial \boldsymbol{z}_1} = \frac{\partial l}{\partial \boldsymbol{z}_2}$. Based on Equation equation 11, we conclude that $\frac{\partial l}{\partial \boldsymbol{x}_2} = \frac{1}{k}\frac{\partial l}{\partial \boldsymbol{x}_1}$.

## B  EXPERIMENTAL DETAILS.

### B.1  VISION TRANSFORMERS ON CIFAR10.

We train from scratch with AdamW optimizer for 100 epochs (50,000 iterations with 100 mini-batch size) in mixed precision. The resolution of an input image is $3 \times 32 \times 32$, and the patch size is 4.

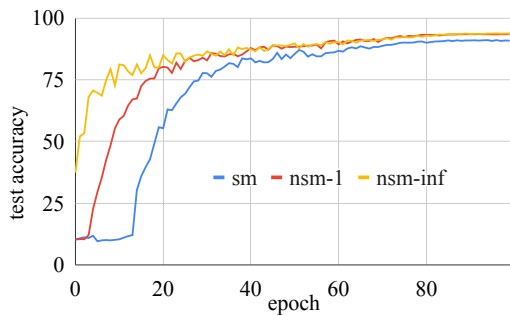

Figure 6: Test accuracy of a ResNet1202 on CIFAR-10

The hidden size, MLP size, number of heads, dimension of heads, and number of layers are 256, 1024, 8, 32, and 8, respectively. We use the following attention function,

$$\text{Attention}(\boldsymbol{Q}, \boldsymbol{K}, \boldsymbol{V}) = \text{softmax}\left(\frac{\boldsymbol{Q}\boldsymbol{K}^T}{\min(\sigma(\boldsymbol{Q}\boldsymbol{K}^T), \gamma)}\right)\boldsymbol{V} \tag{28}$$

where the softmax and the standard deviation are calculated along the same axes. We set $\gamma = \sqrt{d}$ and $+\infty$. The learning rate is 1e-3, and a strong weight decay 1e-1 is applied. The learning rate linearly increases from 2e-4 with 5 warmup epochs and then decays to 0 with the cosine annealing scheduler. We also enable label smoothing (0.1), random erasing with the probability of 0.1, mixup with $\alpha = 0.2$, cutmix with $\alpha = 1.0$, and TrivialAugment. The code is attached in the supplementary material.

### B.2 VISION TRANSFORMERS ON IMAGENET.

We follow the torchvision's reference implementation. The batch size is $512 \times 8 = 4096$. We train from scratch with AdamW optimizer in mixed precision. The learning rate is 3e-3, and the weight decay is 3e-1. The learning rate is linearly increased from $3e-3 \times 0.033$ with 30 warmup epochs and then decays to 0 with the cosine annealing scheduler. We also enable label smoothing (0.1), mixup with $\alpha = 0.2$, cutmix with $\alpha = 1.0$, clipping gradient norm with 1, RandAugment (Cubuk et al., 2020), repeated augmentation with 3 repetitions, exponential moving average for model parameters.

### B.3 RESNET ON IMAGENET.

We follow the example in the JAX framework. We only enable horizontal flip data augmentation. We use SGDM with a mini-batch size of 8,192, a learning rate of 3.2, a momentum of 0.9, and a weight decay of 1e-4. The learning rate is warmup in 5 epochs.

### B.4 MACHINE TRANSLATION.

We use exactly the same settings as Xu et al. (2019). Please refer to the original paper for reference.

## C  EXTENDED EXPERIMENTS

### C.1  TRAINING PERFORMANCE OF A DEEP RESNET

We train a ResNet1202, which is investigated in the original ResNet paper (He et al., 2016). Figure 6 compares the test accuracy during the training process. NormSoftmax can significantly acceler-ate the training process and achieve better test accuracy than the standard softmax. With standard softmax, the model is stuck in the low confidence zone in the initial stage.

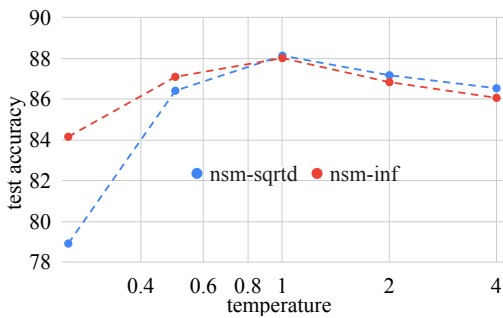

Figure 7: Test accuracy of a ViT with different temperatures after normalization.

## C.2 SCALING FACTORS IN ATTENTION

The default scaling factor in attention is $d^{-1/2}$. We sweep the scaling factors for the standard softmax. We apply the corresponding $\gamma$ in NormSoftmax. We follow the same experiment settings as in Section 4.1.

| scaling factor for sm | 1 | $2d^{-1/2}$ | $d^{-1/2}$ | $d^{-1/2}/2$ | $d^{-1}$ | |
|---|---|---|---|---|---|---|
| test accuracy for sm | 80.98 | 86.38 | 87.96 | 87.97 | 86.98 | |
| $\gamma$ for nsm | 1 | $d^{1/2}/2$ | $d^{1/2}$ | $2d^{1/2}$ | $d$ | $\infty$ |
| test accuracy for nsm | 81.04 | 87.16 | 88.13 | 88.21 | 87.1 | 88.01 |

Table 5: Results for (1) the standard softmax (sm) with different scaling factors and (2) NormSoftmax (nsm) with different $\gamma$

Table 5 shows that scaling factors of $d^{-1/2}/2$ and $d^{-1/2}$ achieve the best performance for the standard softmax, similar to (Lee et al., 2021). NormSoftmax shares the same trend with the standard softmax, achieving the best performance with $\gamma$ of $2d^{1/2}$ and $d^{1/2}$. NormSoftmax achieves better results than the standard softmax with the corresponding pair of scaling factor and $\gamma$.

## C.3 TEMPERATURE AFTER NORMALIZATION IN NORMSOFTMAX

We add a temperature parameter $\tau$ in NormSoftmax.

$$\text{NormSoftmax}(\boldsymbol{x}, \gamma, \tau) = \text{softmax}\left(\frac{\boldsymbol{x}}{\tau \min(\sigma(\boldsymbol{x}), \gamma)}\right) \tag{29}$$

If $\gamma = +\infty$, the input of softmax will always have a standard deviation of $\tau^{-1}$. For a finite $\gamma$, we have $\sigma\left(\frac{\boldsymbol{x}}{\tau \min(\sigma(\boldsymbol{x}), \gamma)}\right) \geq \tau^{-1}$

Figure 7 illustrates the results with different temperature parameters. $\tau = 1$ is the best choice for both `NSM-inf` and `NSM-sqrtd`. When $\tau < 1$, `NSM-inf` is much more stable than `NSM-sqrtd` since the softmax input may have a high variance for `NSM-sqrtd`. On the contrary, when $\tau > 1$, `NSM-sqrtd` is better than `NSM-inf`. The softmax input of `NSM-inf` has a low standard deviation, restricting the performance.

## D COST ANALYSIS OF NORMSOFTMAX

Since we import normalization in the softmax, we inevitably introduce the extra computation and memory cost of NormSoftmax. However, we show that the overhead of the proposed lightweight NormSoftmax is negligible for large machine learning models.

**Memory cost.** Equation 6 indicates that we can preprocess the input vector $x$ and pass it to NormSoftmax. We can fuse the $\gamma$ factor with the ascending layer of NormSoftmax if possible during

inference. For a linear layer $x = Ay + b$, we can always fuse the $\gamma$ with $A$ and $b$, as we do not need to save $\gamma$ during inference. For training, we need to pay the extra memory cost for normalization since we have to save intermediate results for gradient computation.

**Computation cost.** For training, we have to pay the cost of calculating the variance and the corresponding gradients. For inference, we may discard the normalization if the softmax is in the last layer. In a classification model with cross entropy loss, we can discard the normalization since the normalization has no impact on the classification result.

In our experiments on Transformer, NormSoftmax induces $0.2\% - 1\%$ extra execution time. For the experiments for cross-entropy loss, the extra execution time is less than $0.05\%$ since we only add one normalization layer.

