# OpenReview forum: "NormSoftmax: Normalize the Input of Softmax to Accelerate and Stabilize Training"
_ICLR.cc/2023/Conference — Submitted to ICLR 2023_

### Official Review · Reviewer_ax8S · 2022-10-23

**Confidence:** 4
**Clarity, Quality, Novelty And Reproducibility:** Work is novel and clearly presented.
**Correctness:** 4
**Technical Novelty And Significance:** 4
**Empirical Novelty And Significance:** 3
**Recommendation:** 6

**Strength And Weaknesses:**

The method is a very simple and elegant solution to the problem of logit normalization in the attention layer of transformer networks. The method seems to lead to small performance gains in many settings, and some performance loss in other settings. The most striking result in my mind is figure 3 (d) - that hyperparameter transfer, in terms of hidden size/head dimension, is more robust with the softmax norm.

Figures 3a and 3c are interesting in particular. It seems that for a small number of epochs the method is helpful, but for a large number of epochs it may not be (or even is slightly worse than the baseline). It does seem that the method helps with increasing depth. For figure 3 (a), what happens if the normalization is used early but not later? It seems that it may hurt more than be helpful at later training epochs. For figure 3 (c), how was hyperparameter tuning done for large depth? If both methods were tuned well, then the result seems quite significant.

One experiment that would greatly strengthen the paper is the comparison of the method with the 1/d scaling for the attention layers (with no norm-softmax) - and possibly an experiment where gamma is given by d instead of \sqrt{d}. The 1/d scaling has also been brought up as a method for fixing anomalously large softmax inputs in attention layers, and I'm curious about the relationship between the two methods.

Another interesting experiment would be tuning the inverse-temperature of the softmax after normalization, as in https://arxiv.org/abs/2010.07344. It's possible that the normalization scale of 1 is sub-optimal for attention - there may be cases where a slightly larger softmax norm allows for more useful non-linearities in the attention mask.

**Summary Of The Paper:**

In this work the authors attempt to tackle the issue of large softmax inputs, particularly in attention layers of transformers. They propose to normalize the input to the softmax by min(\sigma, \gamma), where \gamma is a constant, and \sigma is the unscaled norm of the input logits. They show that this normalization seems to improve performance on a variety of tasks, and in addition makes hyperparameter transfer (specifically, changing the attention dimension) more robust.

**Summary Of The Review:**

Overall the method is simple and a good attempt at dealing with the large softmax input problem that plagues attention layers in large models. The ability to transfer hyperparameters more easily across geometry is particularly impressive. If some of the details from Figure 3 were more fleshed out, giving stronger results from the method, I would be comfortable switching my review result to an 8 - the paper is almost there.

---

> ### Author Response · Authors · 2022-11-11
> **Author Response to Reviewer as8S**
>
> We appreciate the insightful feedback from the Reviewer, especially the reviews regarding Figure 3. Our responses are as follows.
>
> ### Figure 3a
> Since the NormSoftmax is the intermediate layer of a Transformer model, it is relatively hard to disable it during the training process. A similar example is the layer normalization layer in Transformer. It is difficult to disable the LayerNorm during the training process. We usually keep these normalization layers in the training and inference.
>
> ### Figure 3c
> We use the same training recipes for all the experiments in this figure for a fair comparison. The only difference is the depth of the ViT. The training hyperparameters are in Appendix B.1. We demonstrate that the NormSoftmax can improve the training for deep ViT. We have added experiments in Appendix C.1 on training ResNets with different depths on CIFAR-10 with similar findings.
>
> ### More experiments
> We have added experiments for attention layers in Appendix C.2 to compare different scaling factors in standard softmax and $\gamma$ in NormSoftmax. We have added experiments in Appendix C.3 to compare different temperatures in softmax after normalization.
>
> Thanks again for your reviews. We look forward to hearing from you.

---

> > ### Comment · Reviewer_ax8S · 2022-11-15
> > **Thanks for the updates**
> >
> > The additional experiments are interesting and help flesh out the paper. It is too bad that it is hard to toggle the normalization; that experiment could have been quite informative.

---

> > > ### Author Response · Authors · 2022-11-16
> > > **Thanks for the comments**
> > >
> > > We thank the reviewer for the comments. We wonder if the reviewer has further insightful suggestions on our paper, e.g., new experiments.
> > >
> > > We would really appreciate it if the reviewer could reconsider the recommendation score accordingly.

---

### Official Review · Reviewer_7Jaw · 2022-10-24

**Confidence:** 3
**Correctness:** 2
**Technical Novelty And Significance:** 2
**Empirical Novelty And Significance:** 2
**Recommendation:** 5

**Clarity, Quality, Novelty And Reproducibility:**

Clarity: motivation and method description is clear, but some reasoning may be missing as I explain it in weakness part;

Novelty: Though the form of method is very simple, I think it's a novel way to address the potential issue of softmax.

Reproducibility: I think the the proposed method can be easily reproduced.

**Strength And Weaknesses:**

Strengths: writing is good, I can easily get the main points of this paper.

Weakness:

1) The motivation of this paper seems not convincing. As far as I can see, the motivation of Normsoftmax is based on the reasoning in sec 3.1, that "the variance of softmax input experiences rapid and huge change during training, especially in the initial stage, which explains why training from scratch is difficult.". I agree with the observation that the softmax inputs suffer from high variance during early training (fig 2 shows that),  but it's not the unique property of softmax, the statistics of batch normalization also fluctuate dramatically at early stage (see fig 2(a) in [1]), and these phenomenons can be well interpreted by the findings of [2], that the training trajectory usually starts from a rugged place to a flat local optimum on loss landscape. The main question is: is this feature bad? Can it really slow down the training process or harm the final performance of the trained model? Current paper is lack of justification on this fundamental reasoning.

2) Current experimental results are not sufficient to verify the effectiveness of the proposed method. The form of Normsoftmax is very simple, which I don't regard as a disadvantage, but very convincing experimental results are required to prove its effectiveness. However, the gap between Normsoftmax and its baseline is too minor: VIT on CIFAR10 is not a good setting to show empirical evidence, let alone the performance is not in SOTA level; As for imagenet and NLP dataset, the improvement on training speed and final performance are both subtle.

3)  Cost analysis (sec 3.4) is ambiguous. Fusing $\gamma$ into linear layer is feasible, but computing $\sigma (x/\gamma )$ still requires extra memory/computation cost during inference, right? I suggest conducting thorough ablation study to study the extra cost brought by Normsoftmax.

[1] Yan, Junjie, et al. "Towards Stabilizing Batch Statistics in Backward Propagation of Batch Normalization." ICLR. 2019.

[2] Li, Hao, et al. "Visualizing the loss landscape of neural nets." NIPS (2018).

**Summary Of The Paper:**

This paper propose a normalization method, NormSoftmax, to remedy the optimization difficulty of softmax when using attention model or cross entropy loss function. Their method is easy to implement, its effectiveness is verified on several data tasks including CIFAR10, Imagenet, and three NLP translation datasets.

**Summary Of The Review:**

I think two major weaknesses needed to be solved to improve the quality of the paper: first, justify the reasoning that why Normsoftmax can improve the training of neural network by stabilizing the softmax input; second, provide more convincing empirical evidence to verify the effectiveness.

---

> ### Author Response · Authors · 2022-11-11
> **Author Response to Reviewer 7Jaw**
>
> We appreciate the insightful feedback from the Reviewer. Our responses are as follows.
>
> Our major target is to accelerate and stabilize training, especially for applications where training speed and robustness are critical. Achieving better task performance is not our primary focus.
>
> ### The reason that our NormSoftmax can improve training
> The input of normalization layers experiences rapid and dramatic fluctuations, as shown in [1] and our paper. Such fluctuations will increase the training difficulty and slow down the learning process. Normalizations are widely used to alleviate this issue by smoothing the optimization landscape [2]. Without normalizations, it is possible that we start from a rugged place and take much more steps to find a flat local minimum. We are also probably stuck in a sharp local minimum.
>
> Above all, the dramatic statistics fluctuation in the early stage is the reason instead of the consequence of applying normalizations. Normalizations can smooth this fluctuation and thus accelerate and stabilize the training.
>
> We inherit the properties of the normalizations and apply them to softmax functions. In Section 3.2, we discuss how NormSoftmax smooths the loss landscape. In Section 3.3, we show that NormSoftmax can help models escape the low-confidence zone and accelerate the transition from distraction to concentration in attention.
>
> [1] Yan, Junjie, et al. "Towards stabilizing batch statistics in backward propagation of batch normalization." arXiv preprint arXiv:2001.06838 (2020).
>
> [2] Santurkar, Shibani, et al. "How does batch normalization help optimization?." NeurIPS (2018).
>
> ### Experiments
> Our major claim is that NormSoftmax can accelerate and stabilize training. Improving the final performance is not our primary target. Our experiments successfully verify our major claim. Experiments with ViT on CIFAR10 demonstrate that our method makes the training robust to model architectures and training recipes. Experiments on ImageNet and machine translation show that NormSoftmax can accelerate the training.
>
> ### Cost analysis
> We refactor Section 3.4 and move it to Appendix D due to the page limit. In the case of cross-entropy loss, it is not necessary to compute $\sigma(x/\gamma)$ if the classification result is the only concern, since the temperature has no impact on the classification result. We have added detailed experiment results in terms of extra cost.
>
> Thanks again for your reviews. We are looking forward to further discussions.

---

> > ### Comment · Reviewer_7Jaw · 2022-11-17
> > **Thanks for your response**
> >
> > Thanks for your response. I have no further questions on the motivation of NormSoftmax and the cost analysis.
> >
> > But I'm still not convinced by the experimental results shown in main text. The acceleration effect is meaningful only when the models can be trained to get its best performance with fewer training budget. The acceleration method cannot be regarded as effective if it can only have slight improvement during early training stage with low performance in practice. For me the acceleration effect of NormSoftmax can only be seen in Figure 6 in appendix. In main text, the gap between baseline (sm) and nsm is around 1% in all kinds of settings during early stage, and they both needs total training epochs to approach their best performance without any clear margin.
> >
> > Based on the current experimental results, I'll keep my recommendation score unless more convincing experimental results are provided.

---

> > > ### Author Response · Authors · 2022-11-18
> > > **Thanks for the comments**
> > >
> > > We appreciate the reviewer's comments. The reviewer and us achieve a consensus on the motivation and cost analysis. The only concern is on the experiments demonstrating the effectiveness of acceleration.
> > >
> > > We add the experiments on SwinTransformer-tiny (Swin-T) in Table 2. The results on Swin-T are similar to ViT-B-16. Due to the limited computation resources, we cannot provide other results before the rebuttal deadline of Nov 18. We will keep working in this direction after the rebuttal.
> > >
> > > Thanks again for the reviewer's insightful feedback.

---

### Official Review · Reviewer_syPR · 2022-10-25

**Confidence:** 4
**Correctness:** 3
**Technical Novelty And Significance:** 2
**Empirical Novelty And Significance:** 2
**Recommendation:** 5

**Clarity, Quality, Novelty And Reproducibility:**

This paper is basically easy to follow.
To my knowledge, the method itself seems original and novel, though the method consists of the combination of well known techniques.

The reproducibility of the proposed method is unknown for the current status since the experiments are not conducted only on a limited settings.

**Strength And Weaknesses:**

Strength:
* Motivation of this study is understandable.
* The method is reasonable.
* This paper is well-organized, easy to follow the main points of this paper.


Weaknesses:
* While the effectiveness of the proposed method only shown in the initial stage of the training phase, it is hard to feel that this method is really effective in the actual use case. I understand the effectiveness, but broader impact in the community seems limited.
* The deep normalization may lead the gradient vanishing problem. This paper does not discuss this point. Therefore, it may have a potential drawback when the model becomes much deeper.


**Summary Of The Paper:**

This paper proposes a variant of the softmax function used in many modules of recently developed deep neural networks.
The authors first point out the shortcoming of the softmax function, which is unstable in the initial stage of model training.
The key idea of the proposed method is to borrow the normalization technique, e.g., layer-normalization, for stabilizing the training to incorporate the variance term in softmax computation.
The experiments are conducted on simple image classification and machine translation benchmark datasets.
The results show that the proposed method performed better in the initial training phase stage.


**Summary Of The Review:**

This paper has certain amount of contributions to the community.
However, the advantage of the proposed method is only limited to the initial stage of the training phase.
It would be much better to consider and show a scenario that the proposed method can show much better advantages.

---

> ### Author Response · Authors · 2022-11-11
> **Author Response to Reviewer syPR**
>
> We appreciate the insightful feedback from the Reviewer. Our responses are as follows.
>
> ### Impact and contributions
> The reviewer mentions that our method can accelerate the initial training. Fast training is critical under some circumstances, such as training with a limited budget [1]. Other than that, we would like to stress two advantages of our proposed method.
>
> First, our method stabilizes the training process, making the training robust to hyperparameters (such as model architectures and training recipes). It is usually challenging to train a machine learning model to achieve peak performance, such as Transformer-based models (details in Section 1). Machine learning practitioners have to find suitable settings through trial and error. Similar to other normalizations, our method can simplify this tuning process and empower researchers to reproduce the existing results and explore new models and methods more efficiently.
>
> Second, our method can regularize the model. Our experiments show that our method obtains neutral or slightly better results in the two cases (dot-product attention and cross-entropy loss function). Our method may have a broad impact on these two widely used cases.
>
> [1] Lin, Ji, et al. "On-Device Training Under 256KB Memory." arXiv preprint arXiv:2206.15472 (2022).
>
> ### Problems with deep models
> We agree that deep models are hard to train due to the issue of gradient explosion or vanishing. Our method addresses rather than induces the gradient vanishing problem. Theoretically, we discuss the gradient normalization in Lemmas 1 and 2. Empirically, Figure 3c demonstrates that our method has a significant advantage over the standard softmax for ViT with large depth. We have added experiments in Appendix C.1 on training a ResNet1202 on CIFAR-10. The NormSoftmax can help the model escape the low-confidence zone quickly and improve the training significantly.
>
> ### Reproducibility
> NormSoftmax is an easy-to-use module to replace standard softmax. Also, most of our experiments follow standard benchmarks, whose details are in Appendix B. For customized experiments, we provide our implementation in the supplementary material. We believe that it is simple to reproduce our method.
>
> Thanks again for your reviews. We are looking forward to further discussions.

---

### Official Review · Reviewer_Eoot · 2022-10-26

**Confidence:** 3
**Clarity, Quality, Novelty And Reproducibility:** 1. Motivation for NormSoftMax in intr…
**Correctness:** 3
**Technical Novelty And Significance:** 2
**Empirical Novelty And Significance:** 2
**Recommendation:** 5

**Strength And Weaknesses:**

Strength:
NormSoftMax extends Logit Normalization by introducing centering and a threshold  gamma in the denominator.

Weaknesses:
1. The introduction does not clearly describe the close relationship between Logit Normalization and NormSoftMax. The authors should first mention the drawbacks of Logit Normalization and then motivate NormSoftMax. Note that Logit Normalization is designed to address a similar issue as NormSoftMax. That is, to reduce the dynamics of the inputs to softmax to make the training easier.
2.  I don't get why Logit Normalization is not compared with NormSoftMax  in their experiments. Or NormSoftMax with gamma=infty is reduced to  Logit Normalization. If it is true, I don't see the advantage of  NormSoftMax with finite gamma. Please correct me if I am wrong.


**Summary Of The Paper:**

This paper extends the Logit Normalization technique by centering and introducing  a threshold  gamma in the denominator, which is referred to as NormSoftMax. Experimental results indicate that NormSoftMax works well in a number of DNN models with either croess-entropy loss or attention modules.

**Summary Of The Review:**

I think NormSoftMax need to be properly motivated in introduction by properly introducing  Logit Normalization. The experiments need to be elaborated to demonstrate the advantage of  NormSoftMax  over Logit Normalization.

---

> ### Author Response · Authors · 2022-11-11
> **Author response to Reviewer Eoot**
>
> We appreciate the insightful feedback from the Reviewer. Our responses are as follows.
>
> ### Motivation of our method
> To our best knowledge, Logit Normalization is proposed to calibrate the confidence [1] or improve the test accuracy [2] when the cross-entropy loss function is used. It is not designed to **stabilize and accelerate** the training process. It is neither proposed in the dot-product attention.
>
> In this paper, we notice a significant change in the softmax input in these two cases, which is one of the reasons for optimization and training difficulty. Motivated by the high variance of the softmax input, we propose NormSoftmax to make the training faster and more robust. Hence, we do not start from the problems of Logit Normalization.
>
> [1] Wei, Hongxin, et al. "Mitigating Neural Network Overconfidence with Logit Normalization." arXiv preprint arXiv:2205.09310 (2022).
>
> [2] Kornblith, Simon, et al. "Why do better loss functions lead to less transferable features?" NeurIPS 2021.
>
> ### Logit Normalization
> We have extended Section 3.2 and added experiments in Section 4.2.
>
> The relationship between Logit Normalization and NormSoftmax is shown in Equation (9). NormSoftmax with $\gamma = +\infty$ first shifts the input to zero-mean and then applies Logit Normalization. Hence, NormSoftmax keeps the invariance under translation, while Logit Normalization loses the invariance.
>
> In extended experiments, we compare these two methods directly in the case of the cross-entropy loss function. NormSoftmax achieves better results than Logit Normalization.
>
> Thanks again for your reviews. Please share any insightful work on Logit Normalization we should discuss and compare. Please let us know if you have any questions.

---

> > ### Comment · Reviewer_Eoot · 2022-11-15
> > **Thanks for the update**
> >
> > I now have a better understanding regarding the relationship between NormSoftmax and Logit Normalization, which is not clearly explained in the first draft. I have two more concerns.
> > 1. It is seen from Table 4 that NormSoftmax with gamma=infty does not always perform better than Logit Normalization.
> > Also NormSoftmax with gamma=1 does not always perform better than NormSoftmax with gamma=infty. This implies that the parameter gamma need to be manually tuned to achieve best performance, which might be time-consuming.
> >
> > 2. Even though Logit Normalization was originally not designed for attention module. I still think it is interesting to also implement Logit Normalization for attention module and see if it performs worse than NormSoftmax. The reason is that the two normalization methods are mathematically very close.

---

> > > ### Author Response · Authors · 2022-11-16
> > > **Thanks for the comments**
> > >
> > > We thank the reviewer for the comments. Our responses are listed below.
> > >
> > > ### $\gamma$ needs to be tuned
> > > $\gamma$ is a parameter in our proposed NormSoftmax, which influences the final performance. We do not tune this parameter for most experiments in our paper and follow the default temperature settings (1 in cross-entropy loss, $\sqrt{d}$ in the attention layer). We have also swept this parameter, as shown in Tables 1 and 5. We find that the default settings are almost optimal.
> > >
> > > Further, the temperature is an inherent parameter that needs to be tuned in softmax to achieve the best performance. For example, the scaling factor in the attention layer is an open problem. $\gamma$ plays a similar role to the temperature in softmax and logit normalization.
> > >
> > > Finally, NSM-inf achieves better or neutral results than LogitNorm in most cases. NSM keeps the invariance under translation, while LogitNorm loses.
> > >
> > > ### Experiments of Logit Normalization in attention
> > > We add experiments in Section 4.1 (Table 1 and related discussions). We find that Logit Normalization is even worse than the baseline, demonstrating the importance of shifting to zero-mean and the effectiveness of our methods.
> > >
> > > Thanks again for the detailed comments. We are looking forward to further discussions. We would really appreciate it if the reviewer could consider the recommendation score accordingly.

---

### Author Response · Authors · 2022-11-11
**General response to all reviewers**

We thank all the reviewers for their valuable feedback. We modify and update our manuscript accordingly, with noticeable changes highlighted in the text and summarized below.
* We extend Section 3.2 and add the corresponding proof in Appendix A.2.
* We refactor Section 3.4 and move it to Appendix D due to the page limit.
* We add Logit Normalization as a baseline in Section 4.2.
* We add more experiments in Appendix C.

Thanks again for all the insightful reviews. Please let us know if there is any question.

---

### Comment · Area_Chair_3xdk · 2022-11-15
**Discussion**

Dear reviewers,

Your response to the authors' rebuttal would be highly appreciated.

Kind regards,
Your AC

---

### Decision · Program_Chairs · 2023-01-20

**Decision:**

Reject

**Justification For Why Not Higher Score:**

The reviewers pointed out a lack of comparison to Logit Normalization, which is included in a revised draft by the authors. There were some questions regarding whether there is still a benefit to the method over a complete training run. There was some discussion. Ultimately, the empirical benefits were insufficient to convince the reviewers towards higher ratings.

**Justification For Why Not Lower Score:**

N/A

**Metareview: Summary, Strengths And Weaknesses:**

Ratings: 5/5/5/6.
Confidence: 3/4/3/4.
Recommendation: Reject.

The softmax function is used in both the cross entropy loss and attention module of transformers. The authors propose a new method called NormSoftmax, which centers the input of the softmax, and normalizes its scale. The paper shows that this results in small improvements in loss and robustness across most experiments. Experiments on transformer-based models and convolutional neural networks showed some benefits of the method.

The reviewers pointed out a lack of comparison to Logit Normalization, which is included in a revised draft by the authors. There were some questions regarding whether there is still a benefit to the method over a complete training run. There was some discussion. Ultimately, the empirical benefits were insufficient to convince the reviewers towards higher ratings.